Vehicle detection and classification using an ensemble of EfficientDet and YOLOv8

Lv Caixia 1
Mittal Usha 2
Madaan Vishu 2
Agrawal Prateek prateek061186@gmail.com 2 3
1 Smart City College of Beijing Union University , Beijing , China
2 Computer Science and Engineering, Lovely Professional University , Phagwara , Punjab , India
3 Shree Guru Gobind Singh Tricentenary University , Gurugram , Haryana , India
Pires Ivan Miguel
Electronic publication date: 2024 Aug 13
Publication date: 2024
Volume: 10
Electronic Location ID: e2233
Received 2024 Mar 1; Accepted 2024 Jul 12
Copyright: ©2024 Lv et al.
Copyright year: 2024
Copyright holder: Lv et al.
License: This is an open access article distributed under the terms of the Creative Commons Attribution License, which permits unrestricted use, distribution, reproduction and adaptation in any medium and for any purpose provided that it is properly attributed. For attribution, the original author(s), title, publication source (PeerJ Computer Science) and either DOI or URL of the article must be cited.
License URL: https://creativecommons.org/licenses/by/4.0/

Keywords: Deep learning, Smart city, Sustainable infrastructure, Computer vision, Object detection, Intelligent transport, Sustainable transport, Intelligent traffic management, Thermal imaging, Vehicle detection

Funding: The authors received no funding for this work.

==============================
With the rapid increase in vehicle numbers, efficient traffic management has become a critical challenge for society. Traditional methods of vehicle detection and classification often struggle with the diverse characteristics of vehicles, such as varying shapes, colors, edges, shadows, and textures. To address this, we proposed an innovative ensemble method that combines two state-of-the-art deep learning models i.e., EfficientDet and YOLOv8. The proposed work leverages data from the Forward-Looking Infrared (FLIR) dataset, which provides both thermal and RGB images. To enhance the model performance and to address the class imbalances, we applied several data augmentation techniques. Experimental results demonstrate that the proposed ensemble model achieves a mean average precision (mAP) of 95.5% on thermal images, outperforming the individual performances of EfficientDet and YOLOv8, which achieved mAPs of 92.6% and 89.4% respectively. Additionally, the ensemble model attained an average recall (AR) of 0.93 and an optimal localization recall precision (oLRP) of 0.08 on thermal images. For RGB images, the ensemble model achieved mAP of 93.1%, AR of 0.91, and oLRP of 0.10, consistently surpassing the performance of its constituent models. These findings highlight the effectiveness of proposed ensemble approach in improving vehicle detection and classification. The integration of thermal imaging further enhances detection capabilities under various lighting conditions, making the system robust for real-world applications in intelligent traffic management.

Introduction

Static traffic management regulations on roadways can lead to congestion as traffic volume continues to grow. Estimating traffic density plays a crucial role in intelligent transportation systems by enabling the development of more efficient traffic management strategies. Detection involves two primary processes: classification and localization (He et al., 2015b; Girshick et al., 2014). Localization identifies the positions of vehicles through bounding boxes, while classification predicts the class of vehicles in images or videos (He et al., 2016). Figure 1 shows an example of object detection in which multiple vehicles are detected and classified and shown by bounding boxes.

Figure 1 Object detection example.

The raw image was obtained from the FLIR dataset. FLIR (https://www.flir.com/oem/adas/adas-dataset-agree/) allows anyone to use this dataset for non-commercial research and academic purposes.

Technological advancements, such as the availability of high-quality cameras and faster hardware, have opened the door to sophisticated systems for addressing modern challenges. In traffic management, a key application is optimizing green signals at intersections based on real-time traffic density to reduce congestion and minimize vehicle waiting times. This innovative method uses modern tools to improve the efficiency and responsiveness of traffic signals, leading to smoother and more efficient urban transportation. Inductive-loop technology, used since the early 1960s, detects vehicles with an electronic unit and a wire loop embedded in the road. Traditionally, radars have been key in traffic monitoring, using radio waves to determine a vehicle’s distance, direction, or speed. However, recent advances in imaging technology have introduced new ways to detect vehicles on the road. This shift towards more diverse and advanced methods has the potential to improve the accuracy and capabilities of modern traffic monitoring systems (Mittal, Srivastava & Chawla, 2019b).

In vision-based vehicle detection, algorithms fall into three categories: motion-based, handcrafted feature-based, and CNN-based techniques. Motion-based methods like optical flow, background subtraction, and frame subtraction detect and classify moving vehicles using static camera images and videos based on their dynamic characteristics. Handcrafted feature-based methods such as HOG (Dalal & Triggs, 2005), SIFT (Lowe, 1999), and Haar-like features (Lienhart & Maydt, 2003) rely on low-level feature representation and require expert intervention for feature extraction. These methods work well with moderate datasets and do not need specialized hardware. Deep learning (DL) models use high-level features learned from large training datasets, eliminating the need for handcrafted feature extraction (Dalal & Triggs, 2005). These models are highly efficient but require significant computational power, typically provided by GPUs (Lienhart & Maydt, 2003). This combination of advanced techniques highlights the evolving landscape of vehicle detection technology.

The main challenge in vehicle detection is dealing with different lighting conditions and video quality. While vehicles are easy to identify during the day by recognizing features like the front, rear, edges, and corners, it becomes much harder at night. To address this, our study uses a dataset of thermal images and compares them with visible image datasets to analyze detection capabilities. Previous studies have shown that algorithms often struggle to detect certain vehicles and sometimes label them inaccurately. To overcome these challenges, a hybrid model that improves vehicle detection for various sizes and shapes has been developed. A majority voting scheme is employed to assign the final label to each vehicle, enhancing accuracy and robustness, especially under poor lighting conditions. The key contributions of the study are:

(a) Collection of thermal and visible images from the FLIR dataset.

(b) Image pre-processing to improve the quality of collected images.

(c) Data augmentation to increase the dataset size and address class imbalances, thereby enhancing the model ability to generalize.

(d) Implementing a majority voting classifier for vehicle detection, leveraging an ensemble approach that combines EfficientDet and YOLOv8.

(e) Performance analysis and comparison between the proposed ensemble model and its individual components, assessing their performances to determine the effectiveness of the ensemble approach.

The subsequent sections of the article are structured as follows: ‘Related work’ presents an in-depth literature review, exploring existing research and insights. ‘Proposed methodology’ outlines the proposed approach, detailing the methodology and key components of the model. ‘Results and discussion’ thoroughly discusses the experimental results. Finally, ‘Conclusion’ encapsulates the article with a conclusion, summarizing key findings and potential implications for future research.

Related work

To address the challenges of vehicle recognition and categorization, Zhang, Xu & Feng (2016) employed a deep neural network (DNN) with the primary goal of extracting high-level features from low-level ones. The study demonstrated the superiority of the DNN over a typical neural network, achieving a reduced error rate of 3.34% compared to 6.67% for classifying cars. Harsha & Anne (2016) involved enhancing a Gaussian mixture model (GMM) with background reduction for vehicle detection. Dimensionality reduction was performed using principal component analysis (PCA) and linear discriminant analysis (LDA), with features extracted using AlexNet and SIFT (Scale-Invariant Feature Transform). A support vector machine (SVM) was then employed for classification, resulting in improved precision, especially at the FC6 and FC7 layers. In a different study, Zhou et al. (2016) developed a deep neural network using the AlexNet DNN for classification and incorporated the YOLO (You Only Look Once) approach for detection. The model addressed challenges in vehicle classification within dark images through scene alteration, late fusion, and color transformation methods, enhancing the performance of the DNN. Gao & Lee (2015) presented a system to identify moving cars using frame difference. They employed a symmetrical filter to create a binary frontal view of the car, identifying the car model using a three-layer limited Boltzmann machine. Considering adverse conditions like poor lighting and inclement weather, Chan et al. (2012) proposed a vision-based method for identifying cars on a roadway. Further exploration involved the tracking of solitary objects using thermal images within a brief timespan (Berg, Ahlberg & Felsberg, 2015; Mittal, Srivastava & Chawla, 2019a).

Another study (Rodin et al., 2018) focused on detecting and classifying objects on the sea surface using thermal images captured by unmanned aerial systems (UAS). This work proved valuable for finding and recovering maritime artifacts, achieving a high accuracy of 92.5% on a test dataset. Additionally, a surveillance system (Nam & Nam, 2018) was presented for classifying and recognizing cars both during the day and at night, incorporating various criteria for feature extraction, including textures, entropy, homogeneity, energy, and contrast. For vehicle detection in Unmanned Aerial Vehicle (UAV) photographs, a catalog-based technique was employed (Moranduzzo & Melgani, 2014). The author addressed the limitations of current methods by recognizing asphalted areas as potential regions for car detection, improving accuracy through the recovery of Histogram of Oriented Gradient (HOG) characteristics. A cloud-based intelligent urban video surveillance system for automatic vehicle detection and tracking was also developed (Chen et al., 2013).

Another strategy Tuermer et al. (2013) involved a Heavy Traffic-Aware HOG (HTA-HOG) based on areas with heavy traffic. Hybrid DNN techniques were utilized by the author Prabha & Shah (2016) for item identification and categorization, incorporating non-negative matrix factorization (NMF) for feature extraction and data compression. In the presence of sudden illumination changes and camera shake, an author Chen, Ellis & Velastin (2012) utilized a background GMM and a shadow removal technique for car recognition, tracking, and division into four groups. The Kalman filter was employed for tracking. A deep Convolutional Activation Feature (DeCAF) was created for vehicle recognition and classification (He et al., 2015a), where visual information was extracted and compared across various methods, including deep CNN and large-scale sparse learning.

Based on vehicle structures, Chen, Ellis & Velastin (2011) proposed a classification scheme that involved manual segmentation for boundary extraction and feature extraction. Vijayaraghavan & Laavanya (2019) utilized the Fast region-based convolutional network (Fast R-CNN) for car detection, capturing images with a camera and balancing the dataset among three classes. The model exhibited acceptable accuracy under the assumption of noise-free, clear images. Finally, Ma et al. (2020) leveraged sensor-based data for traffic analysis by strategically placing sensors next to the road and selecting data from diverse sources for further analysis. Byun et al. (2021) presented an innovative approach utilizing a deep neural network for the automatic estimation of vehicle speed from videos captured by unmanned aerial vehicles (UAVs). The proposed method encompassed several key steps: (i) detecting and tracking vehicles through video analysis, (ii) determining image scales by analyzing distances between road lanes, and (iii) estimating vehicle speeds. Remarkably, this approach enables the automatic measurement of vehicle speed solely from UAV-recorded videos without requiring additional information in both directions on the roads simultaneously. The results showcased a robust performance, with a 97.6% recall rate and a 94.7% precision rate in vehicle detection.

Al Mudawi et al (2023) proposed a model structured into five distinct stages. In the initial stage, all images underwent preprocessing to diminish noise and enhance brightness levels. Subsequently, foreground items were extracted from these images through segmentation. The segmented images were then inputted into the YOLOv8 algorithm for vehicle detection and localization in each image. Following this, the detected vehicles underwent feature extraction, encompassing Scale-Invariant Feature Transform (SIFT), Oriented FAST and Rotated BRIEF (ORB), and KAZE features. For classification purposes, the Deep Belief Network (DBN) classifier was employed. The experimental results, spanning three datasets, showcased notable improvements. The proposed model achieved an accuracy of 95.6% on the Vehicle Detection in Aerial Imagery (VEDAI) dataset and 94.6% on the Vehicle Aerial Imagery from a Drone (VAID) dataset, respectively.

Mittal & Chawla (2023) introduced a system that gathered data from diverse open-source libraries, including FLIR, KITTI, and MB7500. The images were annotated to identify vehicles in six distinct classes. To address the challenge of imbalanced datasets, the researchers applied data augmentation techniques. Subsequently, they developed a model utilizing an ensemble of Faster region-based convolutional neural networks (Faster R-CNN) and Single-Shot Detector (SSD), which was trained on the meticulously processed datasets. To evaluate the performance of the proposed model, comparisons were made with base estimators on the FLIR dataset (both thermal and RGB images separately), MB7500, and KITTI datasets. The experimental outcomes revealed that the proposed ensemble achieved the highest mean average precision (mAP) of 94% on the FLIR thermal dataset. This surpassed SSD by 34% and outperformed the Faster R-CNN model by 6%. These findings underscore the effectiveness of the ensemble model in significantly enhancing object detection performance across thermal and RGB images. Table 1 shows a summary of state-of-the-art works published for vehicle detection and classification.

Table 1 Summary of literature review for vehicle detection and classification.

Year	Authors	Image type	Approach	Outcome	
2017	Zhang, Xu & Feng (2016)	RGB images	A neural network with multiple layers was employed.	In contrast to a conventional neural network with a 6.67 percent error rate, the deep neural network exhibited a lower error rate of 3.34 percent.	
2016	Harsha & Anne (2016)	RGB images	Enhanced GMM with background reduction, PCA, LDA, AlexNet, SIFT, SVM	Improved precision, especially at FC6 and FC7 layers	
2016	Zhou et al. (2016)	RGB images	Deep neural network using AlexNet DNN, YOLO approach for detection	Improved vehicle classification in dark images through scene alteration, late fusion, and color transformation methods	
2015	Gao & Lee (2015)	Binary images	Frame difference and symmetrical filters were employed in the process.	Attained a perfect accuracy of 100% on the provided dataset.	
2011	Chan et al. (2012)	CCD camera images	In this approach, a clustering technique was implemented, and a comparative analysis was presented between the AdaBoost classifier and the newly proposed system.	The model underwent evaluation, taking into account seven videos recorded under conditions of low illumination.	
2015	Berg, Ahlberg & Felsberg (2015)	Thermal images	Proposed Thermal Infrared Benchmark for STSO tracking	Introduced a new benchmark with LTIR dataset; Distinctions in tracking principles’ ranking n oted between visual and thermal benchmarks	
2019	Mittal, Srivastava & Chawla (2019a)	Thermal images	Faster R-CNN on Thermal and Visual Spectrum	Thermal images outperform in recall and accuracy, with 75.9% accuracy for 4-Wheelers (vs. 24.3% in visible spectrum). Valuable for challenging traffic monitoring. Future potential for advanced deep learning on both thermal and normal images.	
2018	Rodin et al. (2018)	Thermal images	Utilizing thermal images captured by UAS	Detection and classification of objects on the sea surface, valuable for recovering maritime artifacts, achieving a high accuracy of 92.5% over a test dataset.	
2018	Nam & Nam (2018)	Both visible light and thermal images were considered.	A Gaussian mixture model was employed in the analysis.	The visible spectrum images demonstrated an accuracy of 92.7%, whereas the thermal images exhibited a slightly lower accuracy of 65.8%.	
2014	Moranduzzo & Melgani (2014)	UAV images	A structured-based approach and Support Vector Machine (SVM) were implemented in the methodology.	Increased accuracy was needed to effectively capture a broader range of potential movement directions.	
2013	Tuermer et al. (2013)	Disparity images	In this context, the HOG (Histogram of Oriented Gradients) method was employed for vehicle detection.	The suggested model exhibited both faster processing speed and more precise vehicle detection capabilities.	
2016	Prabha & Shah (2016)	RGB images	Hybrid Deep Neural Network (DNN) with NMF	Utilized for item identification and categorization, incorporating non-negative matrix factorization (NMF) for feature extraction and data compression.	
2012	Chen, Ellis & Velastin (2012)	RGB images	Background GMM, Shadow Removal, Kalman Filter	Used for car recognition, tracking, and division into four groups in the presence of sudden illumination changes and camera shake.	
2015	He et al. (2015a)	RGB images	Deep Convolutional Activation Feature (DeCAF)	Created for vehicle recognition and classification, involving the extraction and comparison of visual information using various methods such as deep CNN and large-scale sparse learning.	
2011	Chen, Ellis & Velastin (2011)	Images were captured from CCTV cameras.	The classification process involved the use of both Support Vector Machine (SVM) and Random Forest classifier.	It attained a classification accuracy of 96.26% for SVM, surpassing the performance of the random forest algorithm.	
2019	Vijayaraghavan & Laavanya (2019)	RGB images	Involved the use of a Fast R-CNN (Region-based Convolutional Neural Network) model.	The model achieved an accuracy of 88% on custom data. A notable limitation of this model is its suboptimal performance when presented with noisy images, yielding results that may not meet acceptable standards.	
2020	Ma et al. (2020)	Data collected from sensors	Sensors were employed to gather vehicle data from roadways, followed by cascade filtering to systematically select and organize the data. Classification of vehicles was carried out using Convolutional Neural Network (CNN) techniques.	The model based on Convolutional Neural Network (CNN) demonstrated a precision level of 98%.	
2021	Byun et al. (2021)	RGB videos	Deep neural network Has been used.	Achieved a Precision of 94.7%.	
2023	Al Mudawi et al. (2023)	RGB Images	YOLOv8, SIFT, ORB and DBN has been utilised.	Resultant accuracy achieved as 95.6% for VEDAI Dataset and 94.6% for VAID.	
2019	Mittal & Chawla (2023)	Thermal images	Ensemble of Faster R CNN and SSD has been used.	Achieved a mAP of 94%	
2024	Chughtai & Jalal (2024)	Various traffic scenarios	DeepLabv3_ResNet101 for vehicle detection and classification	Improved performance in multi-class vehicle detection and classification	
2024	Ismail & Ali (2024)	Videos	Transfer learning with YOLOv5, Mask R-CNN, and SSD for vehicle detection and classification	Improved performance in real-time vehicle detection and classification	

From the literature, we analyzed that there is less exploration of combining multiple models that can leverage the strengths of each model to improve the detection accuracy. Previous studies overlook the pre-processing steps, like noise reduction, contrast enhancement, specific to thermal images which can significantly affect detection performance. Although transfer learning is very common yet there is a lack of focus on fine-tuning pre-trained models particularly for thermal images. Also, there is a need for larger and more diverse datasets that includes a variety of vehicle types and environmental conditions.

Many studies rely on specific datasets that may not cover the full spectrum of real-world scenarios. This can limit the generalizability of the models. Also, there is lack of studies that explores the fusion of thermal images with other types of data like visible spectrum images.

Proposed Methodology

Initially, thermal and RGB images are collected from the Forward-Looking Infrared (FLIR) dataset (https://www.flir.in/oem/adas/adas-dataset-form. The thermal images are captured using infrared cameras. Thermal cameras detect heat emitted by objects and thus can detect the objects in low-light conditions whereas the traditional cameras might fail to do so. The dataset includes a wide range of scenes, including urban environments, highways, and rural areas. This diverse characteristic helps in training the generalized models well across different settings. Given the limited dataset and the challenging conditions under which the images were captured, data augmentation and pre-processing methods have been applied to improve data diversity and enhance image quality. Table 2 shows the original data distribution per class as well data distribution after augmentation. Each image is normalized and resized to 640X640 size. Adaptive histogram equalization method is applied to enhance the contrast of images. Further, images are annotated using LabelImg tool and categorised into six different classes i.e., cycle, two-wheeler, light vehicle, heavy vehicle, bus and truck.

Table 2 The data distribution per class.

Class	Number of images before augmentation	Number of images after augmentation	
Cycle	550	2,600	
Two wheeler	600	2,750	
Light vehicle	2,000	4,500	
Heavy vehicle	2,400	5,200	
Bus	500	2,400	
Truck	450	2,300	

Figure 2 Proposed methodology.

Data augmentation helps to artificially increase the size and diversity of the dataset, addressing class imbalance and improving model performance. Various data augmentation methods such as rotation, scaling, flipping, noise addition etc. are applied. Rotation helps the model to recognize objects from different angles, improving its ability to detect vehicles in various orientations. Scaling helps the model to handle variations in object size and distance, making it robust to changes in scale. Flipping increases the variability of object orientations, helping the model to recognize vehicles regardless of their direction. Noise Addition simulates real-world image imperfections, such as sensor noise or environmental factors, and makes the model robust to noisy data.

Figure 2 shows flow diagram of the proposed methodology. In the presented study, an ensemble approach involves combining the outputs of two deep learning models, namely EfficientDet and YOLOv8. Ensemble learning is a machine learning technique that involves training multiple models, known as base learners or weak learners, and combining their predictions to improve overall performance. This method leverages the diversity of the individual models to reduce variance, bias, and improve the robustness and accuracy of the final predictive model. EfficientDet offers a balanced approach with its BiFPN, providing robust multi-scale detection without the high computational cost of traditional two-stage models. YOLOv8 excels in speed and simplicity, making it ideal for real-time applications. By leveraging these models, a high-performing, efficient vehicle detection and classification is achieved. Both the models are single stage detectors but EfficientDet combines the benefits of single stage speed with multi scale feature fusion normally seen in two-stage detectors.

EfficientDet

EfficientDet is a state-of-the-art object detection model that combines efficiency and accuracy. It was proposed by researchers at Google in 2019. The main idea behind EfficientDet is to use efficient backbone networks, such as EfficientNet, along with a novel compound scaling method to balance model size and accuracy. The architecture is designed to be computationally efficient while achieving competitive performance on object detection tasks. EfficientNet is known for its ability to achieve high accuracy with a relatively small number of parameters by using a compound scaling method that optimizes the depth, width, and resolution of the network. EfficientDet introduces a new type of feature pyramid network called BiFPN. Traditional Feature Pyramid Networks (FPN) combine features from different levels of a backbone network to create a multi-scale feature pyramid. BiFPN extends this idea by introducing bidirectional connections to allow information to flow both up and down the pyramid. The model includes a detection head responsible for predicting bounding box coordinates, class labels, and objectness scores for each anchor box at different scales. EfficientDet uses a compound scaling strategy to balance the trade-off between accuracy and efficiency. Like many object detection models, EfficientDet utilizes anchor boxes at different scales and aspect ratios. The model regresses these anchor boxes to predict the final bounding box coordinates. Figure 3 shows the architecture of EfficientDet model.

Figure 3 Architecture of EfficientDet.

Raw input image source: FLIR dataset.

YOLOv8

The YOLOv8 is the latest version of the YOLO algorithm, surpassing its predecessors through the incorporation of enhancements like spatial attention, feature fusion, and context aggregation modules. These enhancements lead to swifter and more precise object detection, establishing YOLOv8 as a pivotal algorithm in the realm of object detection. The YOLOv8 employs a convolutional neural network divided into two primary components: the backbone and the head. Figure 4 shows the architecture of YOLOv8 model. The backbone of YOLOv8 is constructed using a customized version of the CSPDarknet53 architecture, comprising 53 convolutional layers.

Figure 4 Components of YOLOv8.

This architecture incorporates cross-stage partial connections to enhance information flow among the various layers. The head of YOLOv8 comprises several convolutional layers followed by a sequence of fully connected layers. These layers play a crucial role in forecasting bounding boxes, object-ness scores, and class probabilities for identified objects within an image. A noteworthy characteristic of YOLOv8 is the integration of a self-attention mechanism in the network’s head. This mechanism empowers the model to selectively concentrate on distinct areas of the image, dynamically adjusting the significance of various features based on their relevance to the task at hand. Another noteworthy attribute of YOLOv8 is its proficiency in conducting multi-scaled object detection. To achieve this, the model employs a feature pyramid network designed to identify objects of various sizes and scales within an image.

Proposed ensemble model

Model selection

EfficientDet is known for its scalability and efficiency. It uses a weighted bi-directional feature pyramid network (BiFPN) to fuse features at different scales to detect objects of various sizes. While, YOLOv8 is optimized for speed and accuracy, making it ideal for real-time applications.

Data pre-processing

Before feeding the images into the models, images are pre-procesed to enhance contrast and reduce noise. Various augmentation techniques such as rotation, scaling, and flipping are applied to increase the diversity of the training dataset to improve model robustness.

Ensemble model

Both models are independently trained and fine-tuned for detection. Flow Diagram of the proposed model is shown in Fig. 5. Input images are pre-processed and bounding boxes, class labels, and confidence scores for generated for detected vehicles by both models.

Figure 5 Flow diagram of proposed ensemble model.

The outputs from EfficientDet and YOLOv8 are fused using a non-maximum suppression (NMS) algorithm to combine overlapping bounding boxes and refine detection results. Bounding box with maximum confidence score is retained and other values are discarded.

Results and Discussion

The proposed work implementation is carried out in Python using the Tensorflow API, and the code execution (Mittal & Agrawal, 2024) is facilitated on an NVIDIA GPU with a capacity of 4 GB.

When crafting a deep learning network, critical user parameters include the learning rate, batch size, and network size. Given the absence of a universal rule for batch size determination, an empirical approach is opted. For optimal training of a deep neural network, a high learning rate is advisable with a larger batch size. Conversely, when the batch size is smaller, a lower learning rate is preferred to mitigate the impact of potentially flawed data in each batch.

In the specific context of the EfficientDet model, the linear scaling learning rate method is employed, setting the learning rate to 0.1 in proportion to the batch size ratio of 256. This approach allows for effective calibration of the learning rate based on the batch size. In the experiments, batch size of 32 for model to refine the performance of the model. Similarly, YOLOv8 is trained with learning rate of 0.001, batch size of 64 and number of anchors as 3.

Precision–recall curve (PR curve) and mAP metrices are used to analyze and compare the proposed ensemble model with the EfficientDet and YOLOv8. The PR Curve is valuable for presenting information retrieval outcomes, with a superior curve indicated by a larger area under the curve (AUC). Meanwhile, mAP serves as a metric for evaluating the effectiveness of object detection tasks, involving a comparison between the ground truth bounding box and the detected box. Table 3 presents the calculated mAP values for EfficientDet, YOLOv8, and the proposed model. The highest mAP, reaching 95.5%, is achieved by the proposed ensemble on the FLIR thermal dataset, surpassing EfficientDet by 6.1% and outperforming the YOLOv8 model by 2.9%.

Table 3 Results of EfficientDet, YOLOv8 and proposed ensemble on the FLIR thermal dataset.

Model	Recall	Precision	F1-Score	mAP	
	Thermal	RGB	Thermal	RGB	Thermal	RGB	Thermal	RGB	
EfficientDet	0.977	0.953	0.958	0.937	0.967	0.944	0.894	0.874	
YOLOv8	0.950	0.932	0.932	0.916	0.940	0.923	0.926	0.904	
Proposed ensemble	0.984	0.971	0.969	0.959	0.976	0.964	0.955	0.931	
Notes.

Results for the proposed ensemble are shown in bold.

Figure 6 Precision vs recall curve of EfficientDet, YOLOv8 and the proposed model on FLIR thermal dataset.

Similarly, mAP of proposed model is higher than its base estimators on FLIR RGB dataset. Table 3 also shows the comparative analysis of proposed ensemble model with its base estimator based on precision, recall and F1-score. The proposed model consistently demonstrates superior and more promising results compared to the individual base estimators. Figures 6 and 7 show the precision vs recall curve of EfficientDet, YOLOv8 and proposed model on thermal images and RGB images respectively. Figure 8 shows the comparison of proposed ensemble model with its base estimators based on mAP values on thermal images and RGB images.

Figure 7 Precision vs recall curve of EfficientDet, YOLOv8 and proposed model on the FLIR RGB dataset.

Figure 8 mAP based performance comparison of EfficientDet, YOLOv8 and proposed ensemble based on mAP.

Average precision (AR) is a comprehension metric that summarizes the recall at multiple intersection-over-union (IoU) thresholds. It provides model’s ability to detect objects under various conditions. A higher AR means the model is better at detecting objects. While Optimal Localization Recall Precision (oLRP) evaluates both how well the model detects and how accurately it places the bounding boxes around them. The range of oLRP is 0 to 2 where lower value represents better model. Table 4 shows the comparison of the proposed ensemble with its base estimators based on AR and oLRP. The Table clearly shows the effectiveness of the proposed model compared to EfficientDet and YOLOv8.

Figure 9 shows the detections made by proposed ensemble model. In Fig. 9A, there are actual two vehicles available. The proposed model also predicts two vehicles. Similarly in Fig. 9B, model has predicted one cycle and one heavy vehicle.

Inference time is also very important metric to evaluate the models. On a typical GPU EfficientDet takes around 20 ms per image. Time taken by YOLOv8 is around maximum 10 ms while proposed ensemble model takes around 32 ms. While the ensemble model offers superior accuracy, it inherently involves higher computational complexity due to the simultaneous operation of two deep learning models. In scenarios of traffic management, accuracy is paramount. Thus, additional computational cost may be justified. Table 5 shows the comparison of proposed ensemble method with state-of-the-art-works. Though the existing referred works use different dataset but the used performance metrics serve a relative indicators of model capabilities. This helps in highlighting the strength and potential limitations of various approaches under different conditions. To ensure a fair evaluation, extensive experiments have been conducted on chosen dataset with both thermal and RGB images. The results demonstrate the effectiveness of proposed ensemble approach.

Conclusion

The proposed vehicle detection and classification system presented in this research work demonstrates notable advancements in the field of computer vision and deep learning. This article elaborates and evaluates an ensemble-based deep learning architecture, a collaborative framework merging EfficientDet and YOLOv8 for enhanced vehicle detection. The utilization of majority voting within the ensemble framework significantly enhances the overall predictive performance, contributing to more robust and reliable results. Despite the incorporation of two deep learning models, which inherently increases the model’s time complexity, the resultant boost in overall accuracy is deemed acceptable. However, it is crucial to acknowledge that ongoing efforts are needed to optimize the running time complexity and computational demands of the proposed system. In practical deployments, the ensemble model’s higher computational requirements can be managed by leveraging modern hardware accelerators, such as GPUs or TPUs, and optimizing the inference pipeline. Cloud-based solutions and edge computing can offer flexibility in managing computational loads effectively.

Overall, the proposed work contributes to the continuous evolution of intelligent transportation systems, paving the way for more sophisticated and reliable applications in real-world scenarios. In future, work will be done to validate findings across additional datasets to ensure broader applicability.

Table 4 Comparison of proposed ensemble, EfficientDet and YOLOv8 based on AR and oLRP.

Model	Average recall (AR)	Optimal localization recall precision (oLRP)	
	Thermal	RGB	Thermal	RGB	
EfficientDet	0.89	0.87	0.12	0.14	
YOLOv8	0.91	0.89	0.10	0.12	
Proposed ensemble	0.93	0.91	0.08	0.10	

Figure 9 Detection made by proposed system.

Raw input image source: FLIR dataset.

Table 5 Comparison of proposed method with existing methods.

Technique/Reference	Dataset	Data size	Approach	Accuracy	
Mittal & Chawla (2023)	Visible image (KITTI, MB7500, FLIR)	14,700	Ensemble of faster R-CNN and SSD	92%	
	Thermal images (FLIR Thermal)	9,710		94%	
Nam & Nam (2018)	RGB images	7,438	Gaussian mixture model	92.7%	
	Thermal images	4,452		65.8%	
Rodin et al. (2018)	Thermal images	22,000	Gaussian mixture model	92.5%	
Al Mudawi et al. (2023)	VEDAI	1,200	Fuzzy C Means, YOLOv8 and DBN	95.6%	
	VAID	6,000		94.6%	
Byun et al. (2021)	RGB videos	14,000	Deep neural network	94.7%	
Chen, Ellis & Velastin (2011)	Captured with CCTV	2,055	Model based classification	96.26%	
Proposed ensemble approach	RGB images	15,000	Ensemble of YOLOv8 and EfficientDet	96.4%	
	Thermal images	15,000		97.6%	

We would like to acknowledge the open dataset FLIR images that we used in the proposed work for the research purpose. We also acknowledge Dr. Manik Rakhra (Lovely Professional University, India), Dr. Mohit Kumar (MIT-ADT, India) and Dr. Pawan Kumar Verma (Sharda University, India) for critically reviewing the technical report and giving suggestions in improving the content.

Additional Information and Declarations

Competing Interests

Author Contributions

Data Availability

The authors declare there are no competing interests.

Caixia Lv conceived and designed the experiments, authored or reviewed drafts of the article, and approved the final draft.

Usha Mittal performed the experiments, analyzed the data, performed the computation work, prepared figures and/or tables, authored or reviewed drafts of the article, and approved the final draft.

Vishu Madaan analyzed the data, prepared figures and/or tables, authored or reviewed drafts of the article, and approved the final draft.

Prateek Agrawal conceived and designed the experiments, performed the experiments, performed the computation work, prepared figures and/or tables, authored or reviewed drafts of the article, and approved the final draft.

The following information was supplied regarding data availability:

The Teledyne FLIR Thermal Dataset for Algorithm Training is available at:

https://www.flir.in/oem/adas/adas-dataset-form.

The code is available at Zenodo: Mittal, U., & Agrawal, P. (2024). Vehicle detection on thermal images using Deep learning models. Zenodo. https://doi.org/10.5281/zenodo.11517521.

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
