# Peer review of "Vehicle detection and classification using an ensemble of EfficientDet and YOLOv8"

_PeerJ Computer Science, doi:10.7717/peerj-cs.2233_

## Round 0.1 · original submission · Major Revisions

Based on the reviewers' comments, the manuscript must be revised accordingly.

**Language Note:** PeerJ staff have identified that the English language needs to be improved. When you prepare your next revision, please either (i) have a colleague who is proficient in English and familiar with the subject matter review your manuscript, or (ii) contact a professional editing service to review your manuscript. PeerJ can provide language editing services - you can contact us at [email protected] for pricing (be sure to provide your manuscript number and title). – PeerJ Staff

·

Basic reporting

This article addresses the inherent difficulties AI-based traffic management solutions face in identifying vehicles from thermal images. These difficulties stem from the unique characteristics of thermal images, such as variations in color representation, edges, shadows, and textures, which pose challenges for standard AI models.

Proposed improvements:
1. There appears to be an incomplete sentence starting around line 105. Addressing this and ensuring the overall clarity of the writing would be beneficial.
2.The research gap within the existing literature on vehicle detection using thermal images isn't explicitly addressed in the document. Highlighting the specific gap this research fills would strengthen the overall contribution.
3. Discuss limitations of the study. Discussing limitations strengthens the review by providing a well-rounded assessment. Briefly mention potential limitations identified in the article or based on the review (e.g., specific types of thermal images used, computational cost).
2. Improve figure resolutions.

Experimental design

While the proposed ensemble method combining EfficientDet and YOLOv8 is a valuable contribution, some key aspects require further clarification. The methodology section currently lacks details on how these models are combined.

Proposed improvements:

Describe the ensemble architecture. This is a critical improvement. Understanding how EfficientDet and YOLOv8 are combined (e.g., averaging predictions, weighted combination) is crucial to evaluate the effectiveness of the ensemble method.

Validity of the findings

The results presented in Table 2 suggest a potentially marginal performance gain for the ensemble model. It would be valuable to consider the trade-off between accuracy improvement and the potential computational cost associated with using two models.

The comparison of performance in Table 3 seems to use datasets different from those employed in the authors' work. This renders a direct comparison challenging.

Proposed improvements:
Clarifying results from different datasets. Comparing results directly across different datasets can be misleading.

Reviewer 2 ·

Basic reporting

All comments have been added in detail to the 4th section called additional comments.

Experimental design

All comments have been added in detail to the 4th section called additional comments.

Validity of the findings

All comments have been added in detail to the 4th section called additional comments.

Additional comments

Review Report for PeerJ Computer Science
(Vehicle detection and classification using an ensemble of EfficientDet and YoloV8)

1. Within the scope of the study, vehicle detection and classification studies were carried out with a deep learning-based ensemble model approach using various open source datasets.

2. In the introduction section, vehicle traffic, technological developments, its relationship with deep learning and the difficulties in vehicle detection are basically mentioned. Additionally, the main contributions of the study are clearly stated.

3. In the related work section, deep learning approaches and their outputs in vehicle detection and classification studies based on different image types in the literature are explained. The literature review carried out and highlighted within the scope of the subject is sufficient.

4. Detailed information about the datasets used within the scope of the study should be provided. The initial amount of data for each class, the amount of data for each class after data augmentation, the amount of objects in the classes, the maximum and total number of objects in the images for each class can be added with detailed tables, etc.

5. It should be explained in detail what the data augmentation methods used to eliminate dataset imbalance are and why they are preferred, as well as what the preprocessing stages are.

6. Although there are many different deep learning models that can be used for object detection in the literature, it should be explained more clearly why efficientnet and yolo are preferred. Additionally, this choice should also be interpreted in terms of single-state and two-stage detectors.

7. More detailed information needs to be given about the ensemble model, which is stated to have achieved the highest results. What is the originality point, algorithm, flow chart, etc.?

8. For accurate analysis of the results obtained in the classification and object detection stages, evaluation metrics must be obtained completely. Therefore, find the missing metrics. For example, especially for object detection results, it is recommended to add metrics such as "count of predicted bounding box", "ground-truth bounding box and predicted bounding box in a sample image for each class", "average recall (AR)", "optimal localization recall precision (oLRP)".

9. In order to observe the effect of data preprocessing and data augmentation on classification and detection results, it is recommended to compare the results in the dataset with and without preprocessing and with and without augmentation.

In conclusion; although the purpose of the study and the literature review are clearly stated, attention should be paid to all the above-mentioned parts such as the originality of the model used, model preferences, analysis of the results, and missing metrics.

Cite this review as

---

## Round 0.2 · Minor Revisions

Based on the reviewer comments, the manuscript must have some final very minor revisions.

·

Basic reporting

Revise the manuscript for minor grammatical errors and awkward phrasing to improve readability.

Examples
"efficient traffic management has become a critical societal challenge" change to "efficient traffic management has become a critical challenge for society."
and
"Initially, dataset consisting of Thermal and RGB images are collected from Forward-Looking Infrared (FLIR) dataset. The dataset contains thermal images captured using infrared cameras." - can be simplified to avoid overusing the word 'dataset'

Experimental design

no comment

Validity of the findings

no comment

Reviewer 2 ·

Basic reporting

All comments have been added in detail to the last section.

Experimental design

All comments have been added in detail to the last section.

Validity of the findings

All comments have been added in detail to the last section.

Additional comments

Review Report for PeerJ Computer Science
(Vehicle detection and classification using an ensemble of EfficientDet and YOLOv8)

Thank you for the revision. When the responses to the reviewer comments and the final version of the paper are examined in detail, it is observed that some responses are limited. However, considering both the improvements made in the study and the contribution of the study to the literature, I recommend that this research paper be accepted. I wish the authors success in their future studies. Best regards.

Cite this review as

---

## Round 0.3 · accepted · Accept

Dear authors,

The comments are accurately corrected, and the manuscript can be accepted.